# The Crucial Role of the Interstitial Cells of Cajal in Neurointestinal Diseases

**DOI:** 10.3390/biom13091358

**Published:** 2023-09-07

**Authors:** Egan L. Choi, Negar Taheri, Elijah Tan, Kenjiro Matsumoto, Yujiro Hayashi

**Affiliations:** 1Enteric Neuroscience Program and Department of Physiology and Biomedical Engineering, Mayo Clinic College of Medicine and Science, Guggenheim 10, 200 1st Street SW, Rochester, MN 55905, USA; choi.egan@mayo.edu (E.L.C.); taheri.negar@mayo.edu (N.T.);; 2Gastroenterology Research Unit, Mayo Clinic College of Medicine and Science, Rochester, MN 55905, USA; 3Laboratory of Pathophysiology, Faculty of Pharmaceutical Sciences, Doshisha Women’s College of Liberal Arts, Kyoto 610-0395, Japan; k-matsumoto@dwc.doshisha.ac.jp

**Keywords:** interstitial cells of Cajal, neurointestinal diseases, gastrointestinal motility

## Abstract

Neurointestinal diseases result from dysregulated interactions between the nervous system and the gastrointestinal (GI) tract, leading to conditions such as Hirschsprung’s disease and irritable bowel syndrome. These disorders affect many people, significantly diminishing their quality of life and overall health. Central to GI motility are the interstitial cells of Cajal (ICC), which play a key role in muscle contractions and neuromuscular transmission. This review highlights the role of ICC in neurointestinal diseases, revealing their association with various GI ailments. Understanding the functions of the ICC could lead to innovative perspectives on the modulation of GI motility and introduce new therapeutic paradigms. These insights have the potential to enhance efforts to combat neurointestinal diseases and may lead to interventions that could alleviate or even reverse these conditions.

## 1. Introduction

Neurointestinal diseases comprise a range of medical conditions that result from the interplay and malfunctioning of the nervous system and the GI system [1]. These disorders are characterized by aberrations in the communication among the brain, the enteric nervous system (ENS)—an intricate network of neurons within the GI tract frequently referred to second brain in the gut—and the gut [2]. Notable instances of these diseases include gastroparesis, Hirschsprung’s disease, slow-transit colonic disorders, and various aging-related conditions [1].

These clinical issues are prevalent, with a high incidence rate, significantly impacting patients’ quality of life, physical well-being, and mental health. The manifestation and severity of the symptoms vary depending on the specific type and extent of the disease. The recent coronavirus disease 2019 (COVID-19) pandemic has exacerbated the severity of GI motility disorders in affected patients. This increase is likely attributed to the mental stressors that impact the GI system through the brain–gut axis, thereby highlighting the complex interplay between various diseases and the indirect mechanisms by which they can affect patients [3]. Presently, therapeutic interventions primarily focus on symptom management, aiming to alleviate pain and enhance GI motor function, rather than addressing the underlying pathophysiological mechanisms due to a limited understanding.

The intricate regulation of GI motility involves various components, including the smooth muscle cells responsible for mechanical work; the enteric neurons that establish crucial reflexes as well as control accommodation and the sphincter’s functions; the ICC responsible for generating and propagating the electrical slow-wave activity underlying smooth muscle contractions, and mediating nitrergic and cholinergic neuromuscular neurotransmission; and other regulatory interstitial cells and muscularis macrophages (MMs), a specialized type of immune cell found in the muscular layer of the GI tract [4,5].

The ENS forms a complex network of neurons embedded within the gut wall, orchestrating GI motility through the meticulous coordination of muscle contractions, secretions, and blood flow. ICC, strategically located along the nerve terminals, serve as crucial intermediaries between the autonomic nervous system and the smooth muscle cells, thereby streamlining the transmission of the signals required for regulating muscle contraction. Enteric neurons, upon releasing neurotransmitters such as acetylcholine, incite the ICC to generate electrical slow-wave oscillations, the precursors to muscle contractions [6]. Nitric oxide (NO), a key biomolecule, assumes a central role in this orchestration. Originating from the enteric neurons, NO diffuses to ICC and smooth muscle cells, inducing relaxation and the subsequent expansion of the GI lumen, thereby ensuring synchronized contractions and appropriate relaxation, which are essential for effective digestion and nutrient absorption [7]. It is well established that ENS-associated disorders culminate in a spectrum of GI motility symptoms. A promising therapeutic avenue for these disorders involves replacing the lost neurons via the transplantation of enteric neural stem cells (ENSC). This innovative approach has been shown to rescue impaired motility by promoting the development of neuronal nitric oxide synthase (nNOS, also known as NOS1)-positive neurons, leading to the restoration of nitrergic responses. Remarkably, this therapy also replenished the diminished ICC numbers observed in the nNOS-deficient mouse model [8].

Notably, the dysfunction of the ICC has been linked to various GI disorders, such as delayed stomach emptying, functional bowel disorders, and motility-related conditions [5,9]. Further exploration of the role of the ICC holds the potential to provide valuable insights into the complex regulation of GI motor function and could offer new avenues for understanding and treating neurointestinal diseases. Thus, this review aimed to comprehensively assess the significance of the ICC in the context of neurointestinal diseases, with the ultimate goal of developing novel therapeutic strategies.

## 2. Aging

As individuals age, numerous changes occur in the nervous and GI tract, which can have an impact on the normal functioning of the digestive system [10,11]. Elderly individuals commonly experience various GI problems, including decreased intestinal motility, reduced gastric compliance and accommodation, impaired swallowing, and a weakened lower esophageal sphincter (LES) [10,11]. These age-related alterations in GI motility can lead to symptoms such as indigestion, constipation, diarrhea, and abdominal discomfort [10,11]. These dysfunctions have been associated with early satiety, heightened satiation, and a decline in body weight [12]. Despite being underestimated due to a lack of awareness, overlapping symptoms, diagnostic challenges, and complex mechanisms, these age-associated GI motor dysfunctions significantly affect the individual’s quality of life and increase their vulnerability to conditions such as protein or energy malnutrition, sarcopenia, and frailty [10,13]. Notably, studies have established a correlation between reduced food intake and increased overall mortality rates in both elderly individuals and aged mice [14,15]. In essence, investigating GI motor dysfunction is crucial for improving lifespans and enhancing the quality of life.

Evidence indicates that the ICC reduce with age in the stomachs of humans and experimental animals. A study in humans has shown a decrease of approximately 13% in gastric ICC per decade [16]. Similarly, a decline in gastric ICC has been observed in chronologically aged mice and mice lacking the anti-aging protein α-Klotho (*klotho* mice, a progeria model) [17,18]. The age-related decline in ICC is correlated with gastric dysfunction, specifically impaired fundic relaxation and decreased gastric compliance, rather than delayed gastric emptying [17,18,19]. These changes occur without any loss of enteric neurons in the stomachs of progeric *klotho* mice [18]. The age-associated decline in ICC stem cells or progenitors (ICC-SC) plays a significant role in the loss of ICC and the associated gastric dysfunctions [17]. The decline in ICC-SC with age is linked to the suppression of the extracellular signal-regulated kinase (ERK)1/2 signaling pathway. Reduced cyclin D1 (CCND1) and increased cyclin-dependent kinase inhibitor 2B (CDKN1B, also known as *p27^Kip^*^1^) downregulate the proliferation and self-renewal of ICC-SC by interfering with entry into the cell cycle [17]. Recently, activation of ERK signaling by the nutrient-sensing hormone insulin-like growth factor 1 (IGF1) has been shown to mitigate age-related reductions in ERK phosphorylation, the loss of ICC/ICC-SC, impaired gastric compliance, and reduced food intake [18]. IGF1 appears to improve the overall appearance, increase body weight, and extend the lifespan of *klotho* mice [18].

Unlike the case of those in the stomach, it is still unclear whether intestinal ICC decrease with age. For example, only a slight decline in ICC has been observed in the lower GI tract of *klotho* mice, despite a decrease in enteric neurons [20]. Similarly, a large cohort study found little to no functional changes in the descending colon in humans [21]. Furthermore, the reasons behind the differential behavior of ICC in the upper and lower GI tract are unknown and warrant further investigation.

In summary, the decline in ICC with age is identified as a primary driver of these dysfunctions, particularly in the stomach, and is linked to changes in cell signaling pathways. While interventions targeting the ERK signaling pathway and the proliferation of ICC-SC, such as IGF1, may be valuable in mitigating age-related GI dysfunctions, the role of the ICC in the intestines remains unclear and warrants further investigation.

## 3. Gastroparesis

Gastroparesis is a medical condition that disrupts the normal muscular movement in the stomach, resulting in delayed emptying of its contents into the small intestine [22]. It is a chronic disorder characterized by a prolonged gastric emptying time, compared with the usual rate in a healthy digestive system. In a well-functioning digestive system, the gastric muscular walls contract to facilitate the breakdown of food and propel it forward into the intestines [22]. The most frequently observed symptoms of gastroparesis include nausea, vomiting, abdominal pain, bloating, heartburn, reduced appetite, weight loss, and fluctuating blood glucose levels. While gastroparesis tends to affect women more often than men, the precise reasons for this gender disparity remain incompletely understood [23]. Although the exact cause of gastroparesis is not always clear, certain factors, such as diabetes (particularly Type 1), are known to contribute to its development. The commonly used treatment modalities for gastroparesis include dietary modifications, the use of prokinetic medications such as metoclopramide, and gastric electrical stimulation [22]. The choice of therapeutic options depends on the severity of the symptoms, with the primary objectives being symptom relief, the correction of delayed gastric emptying, and improvements in the quality of life. It is important to note that these current treatments do not provide a cure for gastroparesis, as the exact causes of the condition are not yet fully understood. Thus there remains an unmet need for effective therapies for gastroparesis.

In 2000, Dr. Ordog was the first to discover that the ICC were damaged in streptozotocin-induced diabetic mice, and the reduction in ICC was associated with delayed gastric emptying in these mice [24]. Following this breakthrough, the National Institute of Diabetes and Digestive and Kidney Diseases (NIDDK) established the Gastroparesis Clinical Research Consortium (GpCRC) with the aim of advancing our understanding and treatment of gastroparesis. In 2011, a groundbreaking study conducted by the GpCRC revealed that the ICC were the most significantly affected cell type in patients with both diabetic and idiopathic gastroparesis [25].

The primary cause of the decline in ICC in diabetes was found to be reduced insulin and IGF1 signaling, rather than hyperglycemia [26,27]. The decrease in insulin/IGF1 signaling leads to myopathy through downregulation of the Kit ligand and the ERK signaling pathways [27,28]. Intriguingly, hyperglycemia itself can promote the growth of ICC via the ERK pathway [29,30]. Moreover, the density of ICC is increased, and gastric emptying of solids is accelerated in leptin receptor knock-out (*Lepr^db/db^*) mice, which serve as a mouse model of Type 2 diabetes [30]. These findings in mice can be supported by a clinical study demonstrating rapid gastric emptying in a subset of patients with Type 2 diabetes [31]. A recent study has shown that targeting the genes related to ICC can potentially ameliorate gastric dysmotility [32]. This indicates that the ICC might represent promising therapeutic targets for diabetic gastroparesis.

Another potentially targetable cell type in diabetic gastroparesis is the MMs [33]. The polarization of MMs is a well-established driver of the pathogenesis of diabetic gastroparesis, leading to damage to the ICC [34,35]. During diabetes, MMs undergo proinflammatory polarization, known as the M1 state, instead of anti-inflammatory polarization (the M2 state). These proinflammatory MMs release inflammatory mediators and generate oxidative stress due to the elevated levels of reactive oxygen species (ROS), which adversely affect the viability and function of the ICC [35].

The reduction in insulin and IGF1 signaling, leading to a decline in the ICC and subsequent myopathy, has been identified as a key contributor to diabetic gastroparesis, distinct from the effects of hyperglycemia. This suggests that ICC and MMs, which both undergo changes in diabetes that adversely affect gastric motility, represent promising therapeutic targets for diabetic gastroparesis. Further research is needed to fully understand these relationships and develop targeted therapies that address the root causes of gastroparesis, rather than just managing its symptoms.

## 4. Irritable Bowel Syndrome

IBS is a neurogastrointestinal disorder characterized by abdominal pain, abnormal bowel activity, and changes in the stools’ composition and evacuation from the rectum [33]. The disorder is more prevalent in young adult women and patients with poor psychological states, affecting approximately 5–10% of the general population. There are four subtypes of IBS. IBS-C, or constipation-predominant IBS, is characterized by constipation being the most frequently observed symptom. IBS-D, or diarrhea-predominant IBS, is typified by recurrent loose or watery stools, commonly accompanied by an urgent need to move the bowels and, occasionally, incontinence. Individuals with IBS-D frequently experience abdominal pain and discomfort, which are often alleviated after defecation. IBS-M, or mixed IBS, also known as IBS-A (alternating), is distinguished by alternating episodes of constipation and diarrhea. Individuals with IBS-M typically exhibit the symptoms associated with both IBS-C and IBS-D. Lastly, IBS-U, or unclassified IBS, designates individuals whose symptoms are pertinent to IBS but do not consistently correspond to any single subtype among the other three [36,37]. Apart from purely biological factors such as infection or diet, the severity of IBS symptoms seems to be correlated with external social and emotional factors such as poor relationships and anxiety/depression [36,37]. The pathophysiology of IBS involves various factors, such as evacuation disorders, motor dysfunction, rectal hypersensitivity, idiopathic bile acid diarrhea, carbohydrate malabsorption, barrier dysfunction, and mucosal immune activation [38]. Pharmacological interventions, such as the use of antispasmodics, laxatives, and antidiarrheals, along with lifestyle modifications including dietary adjustments (e.g., consuming psyllium and different types of fiber) and moderate exercise, have all shown potential benefits [39]. Additionally, psychological treatments, such as the use of antidepressants and cognitive-behavioral therapy, have also been used to address the psychological aspects associated with IBS [40]. A recent genome-wide analysis revealed that IBS shared a similar genetic expression of the loci related to mood disorders, anxiety, and/or the nervous system. This finding suggests that these conditions may share pathogenic pathways, rather than having direct causal relationships [41]. As a result, IBS is classified among the disorders of the gut–brain interactions, owing to its connection to the CNS [42]. Regarding the gut microbiome’s role, although it directly influences neuronal pathways and is associated with the microbiota–gut–brain axis, investigations have demonstrated that any correlation between IBS and the gut microbiome’s composition appears to be inconsistent. This inconsistency likely stems from the variable and environmentally dependent nature of the microbiota [43,44,45,46]. However, it is more plausible that the dysfunction of the intestinal barrier associated with IBS leads to microbial infiltration of the mucosa [47]. This, in turn, results in increased mucosal immune activity, as studies have shown that mast cells increase in patients with IBS compared with the controls [48,49], leading to higher secretion of histamine within the intestinal mucosa [50]. A study showed that histamine in patients with IBS sensitized the somatic pain receptors, which explained the pathogenic increase in visceral pain sensitivity seen in most IBS patients [50]. This not only suggests antihistamines as a possible therapy for the painful symptoms of IBS but also creates a connection linking the enteric nervous system, the gut microbiome, and the immune system. Another study indicated that physiological and psychological stress are linked to epithelial permeability in the gut, resulting in increased inflammation and pain in the gut [51]. Mast cells contribute to neuroinflammation and progressive neurodegenerative diseases in the brains of mice [52]. Thus it is quite possible that the same could be said about the effect of mast cells on the ENS. This would lead credence to the study that showed that IBS was associated with a 44% higher risk of Parkinson’s Disease (PD), a chronic degenerative disorder of the CNS [53].

The association between IBS and the ICC can be attributed to several factors. These include the presence of 5-hydroxytryptamine (5-HT, also known as serotonin) receptors on the ICC, an inverse relationship between the ICC and inflammation, and a potential link between the ICC and the visceral hypersensitivity often observed in IBS [54]. Contrary to earlier hypotheses positing a decrease in ICC in IBS patients [55], the relationship between 5-HT and the proliferation of ICC, coupled with the observation that IBS patients exhibit an elevated expression of 5-HT and its receptors (e.g., 5-HT_3_ receptors) within the tissue of the intestinal mucosa, suggests that the ICC may actually increase under the conditions of IBS. The variations in the levels of 5-HT across different IBS subtypes, specifically an increase in IBS–diarrhea and a decrease in IBS–constipation, suggest a complex and nuanced relationship between 5-HT and the ICC. This intricate connection is further emphasized by the observed augmentation of the activity of ICC in response to 5-HT [56,57]. The American Gastroenterological Association Guideline cautions against the use of selective serotonin reuptake inhibitors in IBS patients, a recommendation supported by the findings of clinical trials [58,59]. This guidance aligns with related research demonstrating that mucosal mast cells downregulate serotonin reuptake transporters, leading to increased levels of 5-HT in mice with IBS-D [60]. This finding underscores not only the links among mast cells, increased visceral pain, and ICC through 5-HT but also reveals possible connections between disrupted serotonin reuptake and ICC activity in IBS.

The relationship between IBS and ICC may be analogous to the findings in a study on neuronal loss induced by infectious inflammation. It would be intriguing to investigate whether the presence of MMs could similarly reduce the loss of ICC. Colonic inflammation has been demonstrated to activate neuronal signaling complexes, leading to cell death in myenteric neurons [61,62]. It would be interesting to see if, as in the study just mentioned, the presence of MMs can reduce the loss of ICC as well. Consequently, there may exist a parallel signaling pathway linking inflammation and ICC. Considering the role of ICC as the pacemakers for GI motor function, its loss – attributable to inflammation or other factors – may elucidate the pathophysiological GI motor dysfunction characteristic of IBS [63]. However, given the current scarcity of studies definitively outlining the relationship between ICC and IBS, prioritizing experiments to evaluate the significance of ICC in experimental IBS subjects would be vital. Such efforts could foster further discussions and understanding of the role of ICC in the pathogenesis of IBS.

ICC may actually increase under some IBS conditions due to elevated levels of 5-HT and its receptors, revealing potential connections among disrupted serotonin reuptake, the activity of ICC, and IBS. Further research and experiments evaluating the significance of ICC in experimental IBS subjects are needed to fully understand the underlying mechanisms and develop targeted therapies for IBS.

## 5. Esophageal Achalasia

Esophageal achalasia is an motility disorder of the esophagus characterized by a lack of peristalsis and often a failure of the lower esophageal sphincter to relax. This leads to obstruction of the passage of food and difficulty in swallowing both food and liquids. Common symptoms include dysphagia, noncardiac chest pain, and regurgitation with a pathophysiological tendency towards autoimmune responses [64]. The risk of esophageal achalasia rises with age, with the average age of onset was reported as approximately 66 years [65]. Although esophageal failure is not guaranteed with aging, esophageal motor function does decline, possibly contributing to a higher incidence of achalasia in older individuals [66,67]. While neuronal loss is a frequent cause of achalasia, the precise etiology remains ambiguous. Some evidence suggests that achalasia might have autoimmune origins, akin to PD, indicated by the discovery of Lewy bodies in certain achalasia patients [68]. Further studies indicate potential immune-related causes; for example, disrupted immune cell homeostasis could lead to neuronal loss and subsequent achalasia [69]. A case report highlighted significant eosinophilic infiltration in achalasia, which responded to immunosuppressive therapy [70], and there is a hypothesis that neuronal loss might arise from a prior viral infection, causing sustained damage to the esophageal neurons [71]. A genetic study conducted in 2014 found a robust correlation between an increased risk of achalasia and genetic diversity in HLA-DQ (a cell surface receptor protein found on antigen-presenting cells such as macrophages), hinting at a genetic interplay between achalasia and immune mechanisms [72]. Thus, immunosuppressive treatments should be explored as potential therapies for esophageal achalasia.

Existing evidence has demonstrated the significance of ICC in nitrergic neurotransmission within the fundus [73,74] and other regions of the gastrointestinal tract [75,76,77,78]. However, data from the esophagus suggests that ICC play a more limited role in this function, as illustrated by the distinct occurrence of achalasia in cases of sNOS deficiency, while ICC deficiency resulted in variable degrees of impaired relaxation in the lower sphincter of the esophagus [78,79].

There is a consistent pattern of reduced ICC in achalasia. Often, achalasia patients show elevated levels of mast cells and M1 macrophages in the lower esophagus, which aligns significantly with the observed loss of ICC and neuronal loss [80,81]. Given this correlation, ICC remains an important avenue of investigation in the context of esophageal achalasia, even if its role as a nitrergic intermediary in the esophagus might be diminished.

Despite the limited role of the ICC in nitrergic neurotransmission in the esophagus, the consistent pattern of reduced ICC in achalasia patients, along with elevated levels of mast cells and M1 macrophages, suggests that the ICC remain an important avenue of investigation in esophageal achalasia. Furthermore, there is evidence suggesting a genetic interplay between achalasia and immune mechanisms, but the exact etiology remains unclear, and thus immunosuppressive treatments should be explored.

## 6. Hirschsprung’s Disease

Hirschsprung’s disease is characterized by the absence of the ENS in the rectum of neonates. The absence can span a segment or encompass the entire colon. The disease has a prevalence of 1 in 5000 live births, with a heightened incidence in Asians [82]. Males are more frequently affected than females. Major pathological manifestations includes intestinal obstruction, persistent constipation, and enterocolitis [83]. Surgical resection of the aganglionic segment followed by anastomosis with the healthy bowel remains the primary treatment [84]. Hirschsprung’s disease is genetically linked, commonly associated with mutations in the RET proto-oncogene (RET) gene, which is crucial for enteric neurogenesis [85,86]. This gene is influenced by PAX3 and SOX10 [87] due to their role in the migration and proliferation of nerve crest lineages [88].

Genetic counseling aimed at detecting these genetic anomalies can potentially assist families in understanding their risk, thereby reducing infant mortality.

Infant mortality in Hirschsprung’s disease is often tied to the onset of enterocolitis. The underlying etiology might involve a blend of distal obstruction, immature intestinal barriers, and consequent immune reactions [89]. The linkage between a deficiency in RET and the onset of Hirschsprung’s disease is noteworthy, as RET plays a pivotal role in the development of Peyer’s patches [90]. Dysfunctions in Peyer’s patches can impair IgA and the responses of T cells [91,92,93]. Disruption of endothelin signaling during the developmental stage in the mouse embryo led to a loss of the neural-crest-derived enteric nervous system of the distal colon [94]. Studies on endothelin receptor-B-deficient mice, which are models for Hirschsprung’s disease, have revealed reduced compartmentalization of Peyer’s Patch B cells and diminished IgA secretion [95]. Hirschsprung’s disease appears to induce a shift in M1 macrophages towards a proinflammatory state, leading to an increase in the production of tumor necrosis factor alpha (TNFa). This increased proinflammatory activity could be a factor contributing to the heightened risk of enterocolitis, as well as to the deleterious effects on the ICC [96]. The role of the ICC in the onset of Hirschsprung’s disease is currently a subject of debate. Some investigations have identified a significant decrease in ICC [97,98], while others found no discernible difference from the control groups [99,100].

Given the close interaction of the ICC with the enteric nerves [101], the presence of ICC is presumably vital for restoring the bowels’ motility. Furthermore, Connexin 43, a gap junction protein in the ICC, is noted to have diminished expression in aganglionic sections of Hirschsprung’s disease patients [102,103]. Considering Connexin 43’s significance in intracellular communication [104], a reduction in its levels might suggest compromised communication among the ICC, potentially leading to dysmotility. Previously, a study highlighted the utility of human intestinal organoids as research tools for diseases that are challenging to model in mice [105]. The investigation elucidated the roles of the ENS and ICC in GI motility, providing insights into Hirschsprung’s disease.

While there is currently no approved treatment for Hirschsprung’s disease that directly targets the ICC, experimental approaches, such as stem cell transplantation, gene therapy, and tissue regeneration, take the development and regeneration of the ICC into account, as they are pivotal for the normal functioning of the intestine. Any successful future treatment will likely need to address not only the absence of enteric neurons but also the abnormalities in the network of ICC.

## 7. Parkinson’s Disease

PD is a well-known neurodegenerative disorder characterized by the loss of dopaminergic cells in the substantia nigra (SN) pars compacta and pars cranialis, along with the accumulation of Lewy bodies, composed of abnormal aggregates of alpha-synuclein [106].

Risk factors associated with PD include male gender, advanced age, and specific environmental factors such as exposure to certain pesticides and residing in rural areas [107]. In industrialized nations, the prevalence of PD in the general population stands at 0.3% [108]. The GI symptoms encompass delayed gastric emptying, constipation, hypersalivation, and impaired swallowing [106]. Significantly, a recent study has drawn attention to the early manifestation of GI motor dysfunctions preceding the onset of the motor symptoms related to PD, thereby highlighting the potential significance of these symptoms as early diagnostic markers [109]. It has been demonstrated that PD leads to irregular patterns of the slow gastric waves induced by ICC, resulting in delayed gastric emptying, even in the early stages of the disease. Therefore, electrogastrography (EGG), a noninvasive method for evaluating gastric electric activity, has been considered a potential tool for the early diagnosis of PD [110].

The accumulation of alpha-synuclein deposits in the GI tract has been associated with a damaged neural network and impaired gastric motility [109]. The pioneering work by Braak et al. demonstrated retrograde progression of these depositions to the dorsal motor nucleus of the glossopharyngeal and vagal nerves and in the olfactory nucleus, leading to GI symptoms such as constipation even before the onset of the PD-related motor symptoms and olfactory disturbances [111]. Subsequent studies have detected Lewy’s bodies in the esophageal myenteric plexus, neurons of Auerbach’s and Meissner’s plexus, and the colonic ganglia of PD patients, providing further evidence of neurodegeneration in the GI tract [112]. To investigate the possible impact of ICC on the pathogenesis of PD, salsolinol, a neurotoxin with irreversible effects on SN neurons, was used to induce a PD-like phenotype in mice. Duodenal image analysis confirmed the reduced expression of the receptor tyrosine kinase Kit, a well-established ICC marker, in the salsolinol-injected mice, suggesting the potential direct effect of neurotoxins on both the ICC and neuronal pathways of gastroduodenal reflexes [112]. In a study evaluating the distribution of ICC in PD patients using Kit immunohistochemistry analysis, a significant reduction in ICC was seen in PD patients compared with healthy controls. However, the mechanism underlying the loss of ICC in these patients remains unknown [113]. Contradictory results have emerged from autopsy studies of PD patients, as some studies failed to confirm the deposition of alpha-synuclein in the GI tract or its presence in the ENS but not in the brain [114]. Additionally, another study proposed that the ICC function normally in healthy controls and PD patients, as assessed by the electromagnetic capsule system. This suggests that GI motor dysfunction in PD may be solely attributed to abnormal neurohumoral signals produced by the vagus nerve and myenteric plexus [115]. GI symptoms in PD are complex and likely result from a combination of factors including neurodegeneration, alpha-synuclein accumulation, and dysfunction of the ICC. Further research is necessary to fully comprehend the cause of GI symptoms in PD and to explore potential therapeutic targets for enhancing the quality of life of PD patients [116]. Additionally, factors such as the use of anti-Parkinsonian drugs and long-term laxative use may also influence GI function and require consideration in future investigations [116]. Understanding the mechanistic evolution of ICC in PD patients will undoubtedly shed light on the pathogenesis of GI symptoms in this neurodegenerative disorder. Furthermore, investigating the impact of pathogenic alterations in microbial composition and dysbiosis in the ENS under neurodegeneration holds promise as an area for future exploration [117].

A comprehensive examination of the early diagnostic markers, the accumulation of alpha-synuclein, the impacts of neurotoxins on ICC, the distribution and functionality of ICC, conflicting autopsy reports, neurohumoral signaling, therapeutic targets, impacts of medication, and alterations in the microbial composition and dysbiosis is imperative for individuals with PD. Elucidating the mechanistic alterations of the ICC in PD patients is crucial for gaining a deeper understanding of the origins of GI symptoms in this neurodegenerative disease.

## 8. Multiple Sclerosis

Multiple sclerosis (MS) is an autoimmune and neurodegenerative disease that causes inflammatory damage within the CNS. MS affects both the white matter’s tracts and the cortical and deep gray matter, causing acute and chronic disruption that result in neurologic symptoms and motor disabilities in patients with MS [118]. A diagnosis of MS requires evidence of damage in at least two separate areas of the central nervous system, ruling out other possible diagnoses [119]. Among these patients, constipation, fecal incontinence, and dysphagia are the most commonly observed symptoms [120]. However, the association between GI symptoms in patients with MS and the pathological changes in the ENS remains unclear. A pathomechanistic link between the well-established autoimmune attack on the CNS and the pathology of the ENS in MS was defined by a B cell and antibody-dependent mouse model of MS, as studied in 2017 using immunohistochemistry and electron microscopy across different stages of the disease. Despite this, the study did not evaluate the status of ICC in this degenerative process [121].

A recent study investigated morphological alterations in the ICC and in mice with experimental autoimmune encephalomyelitis (EAE), an animal model for MS. The expression of anoctamin 1 (ANO1; another well-established marker for ICC) was decreased in EAE mice. Notably, these findings were consistent with the previous studies that reported a decline in ICC numbers in EAE mice, which was also evident in the bladders of these mice [122].

In summary, there is a need for more comprehensive studies to investigate the unclear association between GI symptoms and the pathology of the ENS in MS patients. Understanding the pathogenesis of MS, including the role of the ICC and ENS, will provide valuable insights into the disease and may lead to the development of new therapeutic strategies.

## 9. Slow-Transit Constipation

Chronic constipation is a prevalent clinical issue with a high incidence rate, ranking sixth among all major GI indications for ambulatory visits [123]. Slow-transit constipation (STC), a subtype of chronic constipation, is characterized by reduced colonic motility, resulting in the delayed passage of stools through the intestines [124]. In this condition, the contraction of the colonic muscles is inefficient, leading to infrequent bowel movements, hard or lumpy stools, straining during defecation, and a sensation of incomplete evacuation [125]. STC can affect both males and females, although it is more commonly observed in women. Age can also influence the development of STC, with a higher prevalence in middle-aged and older individuals. The current therapeutic approach for STC involves conservative measures, such as lifestyle modifications and dietary changes, as well as medical treatments that include prokinetic agents to enhance colonic motility. In specific cases, a subtotal colectomy may be necessary and beneficial for managing symptoms in a small subset of patients with slow-transit constipation [124].

One significant challenge in managing STC lies in its chronic and complex nature. Treatment responses vary among individuals, and some STC patients may not experience a substantial improvement with the available therapies. In such instances, a multidisciplinary approach involving gastroenterologists, dietitians, and psychologists may be required to provide comprehensive care and optimize the treatment outcomes. Furthermore, further research is needed to gain deeper insights into the underlying causes and develop more effective treatments for STC.

The precise etiology of STC remains incompletely understood, but it likely involves factors related to abnormalities in the neural regulation of colonic muscle contractions. Several investigations conducted on individuals with slow-transit constipation have reported a reduced presence of ICC and enteric neurons [126,127]. However, the exact mechanisms responsible for the decline in ICC and neuronal populations remain unclear. One possible hypothesis suggests that inflammatory MM-induced damage might contribute to the deterioration of ICC [128], although the involvement of other immune cells was not observed in these patients [127]. While enteric neurons have been reported to undergo B-cell lymphoma 2 (Bcl2)-associated apoptotic damage in the context of slow-transit constipation [127], this phenomenon is not observed in the ICC, as they are widely recognized for their high resistance to apoptosis [30,129].

There is a pressing need for a more individualized treatment approach for STC patients, comprehensive research to gain deeper insights into the underlying causes of STC, investigations into the exact mechanisms responsible for the decline in ICC and neuronal populations, investigations into the lack of apoptotic damage in the ICC, and the development of more effective treatments for STC. Understanding the pathogenesis of STC, including the role of the ICC, enteric neurons, and inflammatory MM-induced damage, will provide valuable insights into the disease and may lead to the development of new therapeutic strategies.

## 10. Chronic Intestinal Pseudo-Obstruction

Chronic intestinal pseudo-obstruction (CIPO) is characterized by impaired peristalsis, leading to bowel obstruction in the absence of any discernible mechanical blockade within the intestines. Although uncommon, CIPO frequently presents in children within their first year of life, The lack of effective prokinetic interventions result in a high mortality rate among both pediatric and adult populations, often attributed to subsequent pathophysiologies and autoimmune reactions [130]. These pathophysiologies include nausea, vomiting, constipation, diarrhea, small intestinal bacteria overgrowth, malnutrition, bladder dysfunction, and psychological disturbances [131]. Current therapeutic interventions, albeit suboptimal in efficacy, aim to alleviate the symptomatology and encompass prokinetic agents, antibiotics, enteral nutrient supplementation through liquid diets, and intestinal decompression [131,132]. CIPO is frequently concomitant with other medical conditions, rendering its standalone etiological manifestation a rarity. This coexistence has complicated efforts to unravel its precise etiology. Notably, emerging evidence suggests a potential mitochondrial linkage. For instance, a group of female subjects demonstrated anomalies in the mitochondrial oxidative phosphorylation pathway, which, although atypical, when coinciding with CIPO, culminated in severe nutritional inadequacies and elevated mortality [133].

Another avenue of investigation has been the connection between CIPO and ICC. Comparative studies between CIPO patients and those with other obstructive bowel disorders revealed an absence of Kit-positive ICC in the former, contrasted with its presence in the latter [134]. However, many of these patients exhibited comorbidities such as Crohn’s disease and diabetes. Crohn’s disease and diabetes are known for having similar reductions in ICC [135,136] so there was no clear correlation with CIPO alone in this study. A case study of a 14-year-old boy with long-standing GI problems as well as CIPO showed intact ICC in the muscularis but none in the myenteric and submucosal layers [137]. This aligns seamlessly with the established function of ICC-MY in driving slow-wave propagation in the GI tract [138,139], while the muscular layer’s ICC facilitate enteric motor neurotransmission [140,141]. However, these studies were carried out on CIPO cases in teens or adults, which may be different from CIPO with an onset at birth. A study showed that neonates with CIPO showed delayed development of the ICC [142]. This still suggests that dysfunction of the ICC, especially in the myenteric regions of the GI, contributes to the development of CIPO.

In summary, there is an urgent requirement for additional research to explore the potential connection between mitochondrial dysfunction and CIPO, to scrutinize the specific role of dysfunction of the ICC in the development of CIPO, to investigate the differences in the development of CIPO across different age groups, and to formulate more potent treatments for CIPO. Decoding the exact etiology of CIPO, encompassing the roles of mitochondrial dysfunction and the ICC, will furnish invaluable insights into the disease and may catalyze the development of novel therapeutic strategies.

## 11. Inflammatory Bowel Disease

Inflammatory bowel disease (IBD) is a complex idiopathic disorder characterized by an increasing global prevalence [143]. In the context of IBD, substantial evidence has highlighted not only the reduced levels of ICC but also the structural abnormalities in these cells [144,145]. However, the pathophysiological roles of ICC in the progression of IBD remain ambiguous.

Prior research has revealed the therapeutic potential of intraperitoneally administered ICC-SC, which significantly attenuated inflammation in both acute and chronic models of colitis [146]. These beneficial effects appear to be mediated through prostaglandin E_2_-dependent immunosuppression, a molecule previously shown to modulate the functionality of T cells through intricate signaling pathways [147].

Furthermore, a recent study by Ferreira-Duarte et al. has shed light on the hypothetical role of ICC in angiotensin II-mediated colonic contractions, specifically in the setting of experimental colitis [148]. Additionally, there have been indications of the involvement of the ICC in the biosynthesis of NO within experimental colitis models [149]. This implies a dynamic interaction between the neuronal networks and ICC, potentially mediated by nitric oxide and muscarinic receptors.

Although it is evident that the ICC are vulnerable to damage or depletion across various inflammatory disorders, there is a significant opportunity for further investigations into their mechanistic significance in IBD. Such research could be pivotal in developing innovative, targeted therapies for this increasingly common condition.

## 12. Conclusions

In this study, we examined the crucial role that the ICC play in the pathophysiology of neurogastrointestinal disorders. Despite substantial advancements in ENS research in recent years, the exploration of the engagement of the ICC in neurogastrointestinal diseases remains nascent, and the comprehension of the underlying mechanisms continues to be constrained.

A contemporary investigation disclosed that psychological stress can adversely influence GI function by modulating the activities of the enteric neurons and macrophages [150]. With the escalating prevalence of stress in contemporary society, the nexus between the brain and the gut has attracted increased scholarly interest. Nevertheless, the precise role of the ICC in this complex brain–gut interplay is largely undetermined, constituting a compelling domain for future research.

These inquiries bear the potential to shed light on novel therapeutic targets, thereby facilitating the design of interventions that may delay or even reverse the progression of neurogastrointestinal diseases. As we persist in probing these studies, the information gleaned will contribute to a robust foundation for successive breakthroughs in this evolving field (Figure 1).

## Figures and Tables

**Figure 1 biomolecules-13-01358-f001:**
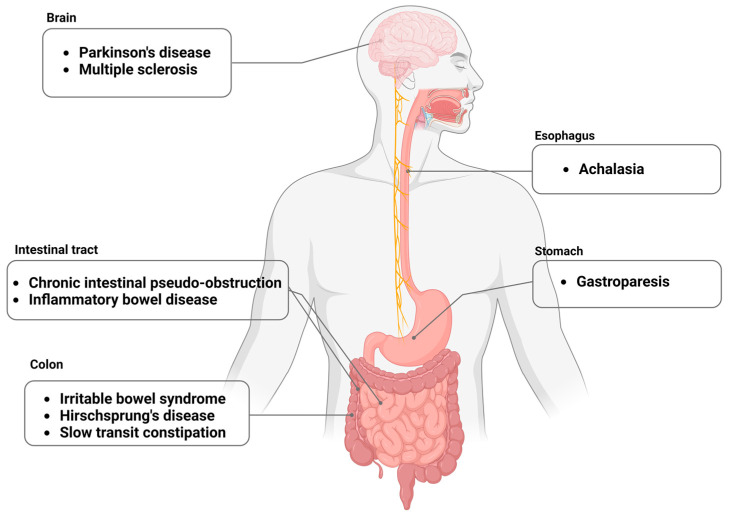
Overview of neurointestinal diseases. ANS, autonomic nervous system; ENS, enteric nervous system.

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
