# Peer review of "The Crucial Role of the Interstitial Cells of Cajal in Neurointestinal Diseases"

_biomolecules, 2023, doi:10.3390/biom13091358_

Round 1

Reviewer 1 Report

Choi and colleagues present a comprehensive review on the possible role of the interstitial cells of Cajal (ICC) in neurointesinal disease. Authors report the presence of ICC deficiencies in several conditions and highlight important areas requiring clarification with future research. It was a great pleasure to read and review this interesting article.

I have a few comments that can serve to improve the manuscript.

1. While the article is comprehensive, it could benefit from adding a paragraph on a few other GI disorders associated with ICC loss, such as inflammatory bowel disease and hypertrophic pyloric stenosis. Colitis is mentioned a few times throughout and warrants its own section.  

2.      It would be nice to have a synthesis of all this literature to make some concluding remarks on the value of the ICC in diagnosis, prognosis, their role in the primary pathology of these conditions, or as a secondary driver of GI sequalae.

3.      In the spirit of the special issue, it would be great to highlight some fundamentals on the functions of the ICC and enteric nervous system, and how they cooperate to regulate intestinal motility. This could include basic anatomy such as their positioning along nerve terminals and the biomolecules involved in signal transduction, such as NO. A study by McCann et al. identified ICC loss in a nNOS deficient mouse model which was reversed by transplantation of neural progenitors. This might further illustrate the importance of the relationships between ICCs and other cell types in the nexus.

McCann CJ, Cooper JE, Natarajan D, Jevans B, Burnett LE, Burns AJ, Thapar N. Transplantation of enteric nervous system stem cells rescues nitric oxide synthase deficient mouse colon. Nat Commun. 2017 Jul 3;8:15937. doi: 10.1038/ncomms15937. PMID: 28671186; PMCID: PMC5500880.           

4.   At the end of page 4 and start of page 5 it reads: Colonic inflammation has been demonstrated to activate neuronal signaling complexes, leading to cell death in myenteric neurons. It would be interesting to see if, like the study just mentioned, MM presence can reduce ICC loss as well. Colitis has been shown to activate neuronal signaling complexes that lead to cell death in myenteric neurons. – this repetition needs to be avoided and the different citations combined.  

Author Response

Reviewer 1

Choi and colleagues present a comprehensive review on the possible role of the interstitial cells of Cajal (ICC) in neurointesinal disease. Authors report the presence of ICC deficiencies in several conditions and highlight important areas requiring clarification with future research. It was a great pleasure to read and review this interesting article.

Response: We greatly appreciate the thorough assessment and meticulous evaluation of our manuscript.

I have a few comments that can serve to improve the manuscript.

  1. While the article is comprehensive, it could benefit from adding a paragraph on a few other GI disorders associated with ICC loss, such as inflammatory bowel disease and hypertrophic pyloric stenosis. Colitis is mentioned a few times throughout and warrants its own section.  

Response: Thank you for your valuable suggestion. We have already incorporated it as a new section, '11. Inflammatory Bowel Disease'

  1.     It would be nice to have a synthesis of all this literature to make some concluding remarks on the value of the ICC in diagnosis, prognosis, their role in the primary pathology of these conditions, or as a secondary driver of GI sequalae.

Response: Thank you for the excellent suggestion. We have incorporated concluding remarks in each section.

  1.     In the spirit of the special issue, it would be great to highlight some fundamentals on the functions of the ICC and enteric nervous system, and how they cooperate to regulate intestinal motility. This could include basic anatomy such as their positioning along nerve terminals and the biomolecules involved in signal transduction, such as NO. A study by McCann et al. identified ICC loss in a nNOS deficient mouse model which was reversed by transplantation of neural progenitors. This might further illustrate the importance of the relationships between ICCs and other cell types in the nexus.

McCann CJ, Cooper JE, Natarajan D, Jevans B, Burnett LE, Burns AJ, Thapar N. Transplantation of enteric nervous system stem cells rescues nitric oxide synthase deficient mouse colon. Nat Commun. 2017 Jul 3;8:15937. doi: 10.1038/ncomms15937. PMID: 28671186; PMCID: PMC5500880.           

Response: Thank you for your valuable suggestion. We have already integrated it into the 'Introduction' section.

  1.  At the end of page 4 and start of page 5 it reads: Colonic inflammation has been demonstrated to activate neuronal signaling complexes, leading to cell death in myenteric neurons. It would be interesting to see if, like the study just mentioned, MM presence can reduce ICC loss as well. Colitis has been shown to activate neuronal signaling complexes that lead to cell death in myenteric neurons. – this repetition needs to be avoided and the different citations combined.  

Response: Thank you for identifying our mistakes. We have omitted the second sentence.

Reviewer 2 Report

Overall comments:

This review discusses how the Interstitial cells of Cajal contribute to various gastrointestinal diseases. The focus is on GI disorders that are neurally mediated, and the authors cover a broad range of conditions involving the enteric and central nervous systems. The title infers that the article will discuss the implications for therapeutics targeting ICC, and I believe this is not really covered in the piece. I suggest the authors adjust the title or the content accordingly to cover this. The functional evidence for the roles of ICC in these diseases could be stronger. Much of the evidence presented highlights the presence or absence of ICC in various diseases, this connection back to physiological changes (where possible) would make the article more convincing. Overall, it is well-written and clear.

Specific comments

Abstract

“Factors like psychological stress, which can negatively affect GI function, make ICC's contribution even more significant.”

I’m not sure what is meant by this sentence.

Introduction

“as the "second brain," and the gut” Change to the ‘second brain in the gut’

The manifestation and severity of symptoms vary depending on the specific type and extent of the disease. The recent Coronavirus disease 2019 (COVID-19) pandemic has led to a surge in GI motility disorders.

The cited paper analyses the incidence of GI symptoms in patients who already have FD or motility disorders during the pandemic. Rather than the onset of a new disease.

The intricate regulation of GI motility involves various components, including smooth muscle cells responsible for mechanical work, enteric neurons that establish cru-cial reflexes as well as control accommodation and sphincter functions, ICC responsible for generating propagating electrical slow wave activity underlying smooth muscle con-tractions, and mediating nitrergic and cholinergic neuromuscular neurotransmission, along with other regulatory interstitial cells and immune/inflammatory cells4, 5.

The referenced articles don't mention the immune cells (assume mean muscularis macrophages).

Aging

  significantly affect the individual's quality of life and increase vulner-ability to conditions like protein-energy malnutrition, sarcopenia, and frailty7, 10. Notably, studies have established a correlation between reduced food intake and increased overall mortality rates in both elderly individuals and aged mice11. Therefore, it is evident that diminished food intake resulting from GI motor dysfunctions may be linked to sarcope-nia, frailty, and ultimately higher mortality rates11, 12.

I’ve underlined the repetition between sentences.

Similarly, a large cohort study found little to no functional changes in the lower GI tract in humans18.

The referenced article did find differences in the ascending colon (which is the lower GI tract).

At the end of the gastroparesis section, muscularis macrophages are mentioned. Given there are articles investigating the role of MM in ageing should they not also be mentioned in this section?

. 2023;2(2):261-276. doi: 10.1016/j.gastha.2022.09.006. Epub 2022 Sep 29.

Age-dependent Microglial Disease Phenotype Results in Functional Decline in Gut Macrophages

Estelle Spear Bishop 1, Hong Namkoong 1, Laure Aurelian 2 3, Madison McCarthy 4, Pratima Nallagatla 5, Wenyu Zhou 5 6, Leila Neshatian 1, Brooke Gurland 4, Aida Habtezion 1, Laren Becker 1

Gut. 2018 May;67(5):827-836. doi: 10.1136/gutjnl-2016-312940. Epub 2017 Feb 21.

Age-dependent shift in macrophage polarisation causes inflammation-mediated degeneration of enteric nervous system

Laren Becker 1, Linh Nguyen 1, Jaspreet Gill 1, Subhash Kulkarni 2, Pankaj Jay Pasricha 2, Aida Habtezion 1

Gastroparesis

At the end of the gastroparesis section, the role of muscularis macrophages affecting the function of ICCs is discussed. Given MMs have demonstrated roles in modulating GI motility, should they not also be discussed here?

Irritable Bowel Syndrome

For clarity describing the difference between the 4 types of IBS would be helpful, and relating the findings of the discussed studies to these phenotypes may provide more context for the role of ICCs in IBS.

“Pharmaco-logical interventions, such as the use of opioids or antagonist/agonist medication…”

The phrase antagonist/agonist really doesn’t mean much here. Could you expand to explain?

“Furthermore, evidence of a connection has been found in a cohort of IBS patients harboring SCN5A mutation which encode for a voltage gated channel on ICC that were not present in IBS-negative controls. These mutations demonstrated voltage-gate abnormalities and may play a role in certain GI motility issues related to IBS52.”

Need to be careful with the wording here as the referenced paper shows channelopathy in some IBS patients (N=6) but does not specifically link this to ICC in these patients. A separate publication by the group links SCN5 to ICC.

HSCR

“This enhancement in proinflammatory activity may be responsible for the elevated risk of enterocolitis, through its adverse effects on ICC 94.”

Can you explain the connection between the proinflammatory activity in HAEC and the link to ICC? It is not clear.

There is quite a bit of literature on the role of ICC in HSCR. A more active discussion on the role of ICC in HSCR, what this might mean for current treatments (ie surgery) and the use of planned therapies (ie enteric regeneration) would be interesting.

“Considering Connexin43's significance in intracellular communication102, its reduced levels might suggest compromised communication among ICC and between ICC and enteric neurons…”

ICC and enteric neurons are not electrically coupled as this sentence suggests. I recommend rewording for clarity.

NA

Author Response

This review discusses how the Interstitial cells of Cajal contribute to various gastrointestinal diseases. The focus is on GI disorders that are neurally mediated, and the authors cover a broad range of conditions involving the enteric and central nervous systems. The title infers that the article will discuss the implications for therapeutics targeting ICC, and I believe this is not really covered in the piece. I suggest the authors adjust the title or the content accordingly to cover this. The functional evidence for the roles of ICC in these diseases could be stronger. Much of the evidence presented highlights the presence or absence of ICC in various diseases, this connection back to physiological changes (where possible) would make the article more convincing. Overall, it is well-written and clear.

Response: We sincerely appreciate the detailed assessment and careful evaluation of our manuscript. As per the suggestion, we have revised the title to 'The Crucial Role of Interstitial Cells of Cajal in Neurointestinal Diseases'. We believe this title more accurately encapsulates the content and focus of our review." 

Specific comments

Abstract

“Factors like psychological stress, which can negatively affect GI function, make ICC's contribution even more significant.”

I’m not sure what is meant by this sentence.

Response: Thank you for your feedback. For clarity, we have removed the mentioned sentences.

Introduction

“as the "second brain," and the gut” Change to the ‘second brain in the gut’.

Response: The necessary modifications have been implemented. 

The manifestation and severity of symptoms vary depending on the specific type and extent of the disease. The recent Coronavirus disease 2019 (COVID-19) pandemic has led to a surge in GI motility disorders.

The cited paper analyses the incidence of GI symptoms in patients who already have FD or motility disorders during the pandemic. Rather than the onset of a new disease.

Response: Thank you for pointing out the mistake. We have amended the sentence to read. “Coronavirus disease 2019 (COVID-19) pandemic has exacerbated the severity of GI motility disorders in affected patients”

The intricate regulation of GI motility involves various components, including smooth muscle cells responsible for mechanical work, enteric neurons that establish cru-cial reflexes as well as control accommodation and sphincter functions, ICC responsible for generating propagating electrical slow wave activity underlying smooth muscle con-tractions, and mediating nitrergic and cholinergic neuromuscular neurotransmission, along with other regulatory interstitial cells and immune/inflammatory cells4, 5.

The referenced articles don't mention the immune cells (assume mean muscularis macrophages).

Response; Thank you for your feedback. For clarity, we have revised the sentence as follows: “with other regulatory interstitial cells and muscularis macrophages (MMs), a specialized type of immune cell found in the muscular layer of the GI tract”

Aging

  significantly affect the individual's quality of life and increase vulner-ability to conditions like protein-energy malnutrition, sarcopenia, and frailty7, 10. Notably, studies have established a correlation between reduced food intake and increased overall mortality rates in both elderly individuals and aged mice11. Therefore, it is evident that diminished food intake resulting from GI motor dysfunctions may be linked to sarcope-nia, frailty, and ultimately higher mortality rates11, 12.

I’ve underlined the repetition between sentences.

Response: Thank you for identifying our mistakes. We have omitted the second sentence.

Similarly, a large cohort study found little to no functional changes in the lower GI tract in humans18.

The referenced article did find differences in the ascending colon (which is the lower GI tract).

Response: Thank you for pointing out our error. We have amended the sentence to read. “Similarly, a large cohort study found little to no functional changes in the descending colon in humans”

At the end of the gastroparesis section, muscularis macrophages are mentioned. Given there are articles investigating the role of MM in ageing should they not also be mentioned in this section?

. 2023;2(2):261-276. doi: 10.1016/j.gastha.2022.09.006. Epub 2022 Sep 29.

Age-dependent Microglial Disease Phenotype Results in Functional Decline in Gut Macrophages

Estelle Spear Bishop 1, Hong Namkoong 1, Laure Aurelian 2 3, Madison McCarthy 4, Pratima Nallagatla 5, Wenyu Zhou 5 6, Leila Neshatian 1, Brooke Gurland 4, Aida Habtezion 1, Laren Becker 1

Gut. 2018 May;67(5):827-836. doi: 10.1136/gutjnl-2016-312940. Epub 2017 Feb 21.

Age-dependent shift in macrophage polarisation causes inflammation-mediated degeneration of enteric nervous system

Laren Becker 1, Linh Nguyen 1, Jaspreet Gill 1, Subhash Kulkarni 2, Pankaj Jay Pasricha 2, Aida Habtezion 1

Response: Thank you for the excellent suggestion. Although the papers mentioned are intriguing, their primary focus is on enteric neurons rather than interstitial cells of Cajal (ICC). Our review is specifically centered on the role of ICC, not enteric neurons, in neurogastrointestinal diseases. Additionally, the guest editors have specifically invited us to write on 'the role of ICC in neurogastrointestinal diseases.'

Gastroparesis

At the end of the gastroparesis section, the role of muscularis macrophages affecting the function of ICCs is discussed. Given MMs have demonstrated roles in modulating GI motility, should they not also be discussed here?

Response: Thank you for your suggestion. While it has been hypothesized that MMs are involved in GI motility, particularly in the small intestine and colon, this section is dedicated exclusively to the examination of gastric motility and ICC. A thorough investigation of the role of MMs in the overall GI motility is beyond the scope of this section.

Irritable Bowel Syndrome

For clarity describing the difference between the 4 types of IBS would be helpful, and relating the findings of the discussed studies to these phenotypes may provide more context for the role of ICCs in IBS.

Response: Thank you for your excellent suggestion. We have outlined the four types of IBS in Section 4.

“Pharmaco-logical interventions, such as the use of opioids or antagonist/agonist medication…”

The phrase antagonist/agonist really doesn’t mean much here. Could you expand to explain?

Response: Thank you for bringing that to our attention. To clarify, we have revised the sentence as follows: "Pharmacological interventions, such as antispasmodics, laxatives, and antidiarrheals.”

“Furthermore, evidence of a connection has been found in a cohort of IBS patients harboring SCN5A mutation which encode for a voltage gated channel on ICC that were not present in IBS-negative controls. These mutations demonstrated voltage-gate abnormalities and may play a role in certain GI motility issues related to IBS52.”

Need to be careful with the wording here as the referenced paper shows channelopathy in some IBS patients (N=6) but does not specifically link this to ICC in these patients. A separate publication by the group links SCN5 to ICC.

Response: Thank you for highlighting this issue. We have removed the mentioned sentences to eliminate any ambiguity.

HSCR

“This enhancement in proinflammatory activity may be responsible for the elevated risk of enterocolitis, through its adverse effects on ICC 94.”

Can you explain the connection between the proinflammatory activity in HAEC and the link to ICC? It is not clear.

Response: Thank you for your comment. The activation of M1 muscularis macrophage-associated pro-inflammatory cytokines in Hirschsprung's disease may indeed lead to colonic inflammation. For better clarity, we have rephrased the sentences as follows: "This increased pro-inflammatory activity could be a contributing factor to the heightened risk of enterocolitis as well as to the deleterious effects on ICC."

There is quite a bit of literature on the role of ICC in HSCR. A more active discussion on the role of ICC in HSCR, what this might mean for current treatments (ie surgery) and the use of planned therapies (ie enteric regeneration) would be interesting.

Response: Thank you for your insightful recommendations. Incorporating future therapies for Hirschsprung's disease would indeed be a valuable addition to this review article. We have also included concluding remarks at the end of this section, as also suggested by Reviewer 1.

“Considering Connexin43's significance in intracellular communication102, its reduced levels might suggest compromised communication among ICC and between ICC and enteric neurons…”

ICC and enteric neurons are not electrically coupled as this sentence suggests. I recommend rewording for clarity.

Response: We have omitted the phrase "and between ICC and enteric neurons". 
